**Data Availability Statement:** We have now provided direct, non-author contact information

# Evaluation of an interprofessional follow-up intervention among people with type 2 diabetes in primary care—A randomized controlled trial with embedded qualitative interviews

**Marit Graue**[1]*, **Jannicke Igland**[1,2], **Anne Haugstvedt**[1], **Ingvild Hernar**[1,3], **Kåre I. Birkeland**[4], **Vibeke Zoffmann**[5,6], **David A. Richards**[1], **Beate-Christin Hope Kolltveit**[1,7]

**1** Department of Health and Caring Sciences, Western Norway University of Applied Sciences, Bergen, Norway, **2** Department Global Public Health and Primary Care, University of Bergen, Bergen, Norway, **3** Department of Internal Medicine, Haukeland University Hospital, Bergen, Norway, **4** Institute of Clinical Medicine, University of Oslo, Oslo, Norway, **5** Interdisciplinary Research Unit of Women's, Children's and Families' Health, Julie Marie Centre, Rigshospitalet, Copenhagen, Denmark, **6** Institute of Public Health Copenhagen University, Copenhagen, Denmark, **7** Vossevangen Medical Center, Voss, Norway

* marit.graue@hvl.no

## Abstract

With an ageing population and improved treatments people live longer with their chronic diseases, and primary care clinics face more costly and difficult-to-treat multimorbid patients. To meet these challenges, current guidelines for the management of type 2 diabetes suggest that an interprofessional team should collaborate to enhance the delivery of worthwhile self-management support interventions. In this study, we aimed to evaluate the effects of an empowerment-based interprofessional follow-up intervention in people with type 2 diabetes in primary care on patient-reported outcomes, biomarkers and weight, and to explore the experiences of patients attending the intervention. We invited patients during regular visits to their general practitioners. The 12-month intervention included 1) empowerment-based counselling; 2) a standardized medical report. The control group received consultations with physicians only. The primary outcome was the Patient Activation Measure, a patient-reported measure assessing individual knowledge, skills, and confidence integral to managing one's health and healthcare. After the trial we conducted qualitative interviews. We observed no difference in the primary outcome scores. On secondary outcomes we found a significant between-group intervention effect in favor of the intervention group, with mean differences in glycemic control after 12 months (B [95% CI] = -8.6 [-17.1, -0.1] mmol/l; p = 0.045), and significant within-group changes of weight (B [95% CI] = -1.8 kg [-3.3, -0.3]; p = 0.02) and waist circumference (B [95% CI] = -3.9 cm [-7.3, -0.6]; p = 0.02). The qualitative data showed that the intervention opened patients' eyes for reflections and greater awareness, but they needed time to take on actions. The patients emphasized that the intervention gave rise to other insights and a greater understanding of their health challenges. We suggest testing the intervention among patients with larger disease burden and a more expressed motivation for change.

from the Regional Committee for Medical and Health Research Ethics South-East Norway through which interested researchers can send data access requests: "Due to personal data protection legislation and legal restrictions related to confidentiality, the data cannot be deposited online as the study participants have not explicitly been informed about, nor approved data sharing when the data were gathered in 2019-2021 (see approval from The Regional Committee for Medical and Health Research Ethics South-East Norway (2019/28/REK south-east A, and the processing of personal data from the Norwegian Centre for Research Data (NSD/ID:821994)). Therefore, application for data use need to be forwarded to the Regional Committee for Medical and Health Research Ethics South-East Norway data due to ethical regulatory conditions of data usage (https://rekportalen.no/#hjem/home).

**Funding:** This study was supported by a post-doctoral position from the Norwegian Nurse Association (BCHK), and the Western Norway University of Applied Sciences. The funders had no role in study design, data collection and analysis, decision to publish, or preparation of the manuscript.

**Competing interests:** KIB has received research support from Astra Zeneca, Bayer, Boehringer Ingelheim, Lilly, MSD, Novo Nordisk, Roche, Sanofi and Sysmex Norway, AH has received honoraria for speaking at meetings from Novo Nordisk and Abbot. IH has received honoraria for speaking at meetings from Novo Nordisk. The other authors have no relevant competing interest to disclose. This does not alter our adherence to PLOS ONE policies on sharing data and materials.

## Introduction

In 2021, diabetes was estimated to affect 536.6 million adults worldwide, with an estimated rise to 783.2 million in 2045 [1]. Accordingly, the health expenditure estimated to be 966 billion USD in 2021 is expected to reach 1,054 billion USD in 2045. The fact that costs are notably related to clinical complications and that complications are significantly associated with glycemia [2] accentuate the urgent need for a structured and targeted follow-up. As advocated by the American Diabetes Association and the European Association for the Study of Diabetes, diabetes self-management support is as important in maintaining recommended glycemic levels as the selection of pharmacotherapy [3]. Self-management support interventions are shown to be capable of lowering HbA1c by at least 0.4% (4.4 mmol/mol), which corresponds to the effects of many glucose-lowering medications [4]. A meta-review of qualitative systematic reviews concludes that a range of self-management support approaches improves clinical and psychological outcomes in people with type 2 diabetes (T2D) [5]. The person living with the condition needs a holistic and multifactorial management approach involving multiple disciplines with specialized clinical knowledge in diabetes and behavior change principles [6].

With an ageing population and improved treatments people live longer with their chronic diseases, and primary care clinics face more costly and difficult-to-treat multimorbid patients [7]. To meet these challenges, current guidelines for the management of T2D suggest that an interprofessional team should collaborate in selecting the glucose-lowering medication, as well as focusing on lifestyle and quality of life issues [3]. A systematic review of the impact of collaboration between physicians and nurses in primary care on patient outcomes conclude that more integrated care models with sufficiently educated nurses may have a positive impact on a number of patient outcomes [8]. Thus, to address future challenges implementing new models of care is needed to optimize the use of existing resources, including more efficient interprofessional collaboration. Instead of a "task shift" between primary care physicians and nurses, a better description of individual health professionals' roles, tasks, and responsibilities of individual caretakes needs to be addressed. In a qualitative study conducted in Norwegian primary care, it is emphasized that although the healthcare professionals perceived to have complementary roles, they did not take full advantage of the potential of sharing care accessible within a team-based approach [9]. In this study, we aimed to evaluate the effects of an empowerment-based interprofessional follow-up intervention in people with T2D in primary care on patient-reported outcomes, biomarkers and weight, and to explore the experiences of patients attending the intervention.

## Material and methods

We undertook a randomized controlled trial (RCT) among people with T2D to compare the intervention with standard care. We used the CONSORT 2010 guidelines for reporting the trial [10]. Ten physicians and six registered nurses delivered the intervention in four primary care clinics in the Western and Eastern parts of Norway. We submitted the trial in Clinical-Trials.gov (ID: NCT04076384) on August 26th, 2019. We intended to complete the intervention in 2019–2020, but because of logistic problems the enrollment of patients to the intervention was from August 27th, 2019, to April 15th, 2020, and the follow-up consultations with three GSD consultations were then conducted from December 1st, 2019, until end of study June 15th, 2021. After conducting the 12-month trial, we conducted individual qualitative interviews with participants from all sites.

We invited potential participants from those with T2D (identified by the diagnostic criteria for diabetes diagnosis (HbA1c ≥48 mmol /mol)) who had participated in a cross-sectional survey conducted in the waiting area among adults aged 20–80 years scheduled for regular consultations at the four clinics from May 2019 to December 2019 (n = 128) [11]. In the survey,

exclusion criteria were severe co-morbidity (severe cancer, severe heart disease, end stage renal disease), major psychiatric disorder (severe depression, bipolar disorder, schizophrenia), recorded cognitive deficiency, and pregnancy. In addition, we excluded people who could not be reached because of logistic or organizational problems or were diagnosed with type 1 diabetes in this intervention trial (n = 23). Among the identified people with T2D, 79 consented to participate. However, three patients withdrew consent after randomization, leaving 40 people in the intervention group and 36 in the control group (Fig 1).

Our pre-study sample size calculation was done using the power-command in Stata. The analysis was based on the 13-item Patient Activation Measure (PAM-13) to detect a treatment effect of 7 points (0.5 SD) (with 80% power and a two-sided 0.05 significance level) yielded a sample size of 64 participants in each trial arm. To account for possible dropouts, we assumed an increase to 77 (20%) in each group to be appropriate. Our initial sample was not sufficient to fulfil this number and accordingly we recruited additional people scheduled for consultations in the four clinics to take part in the intervention. Because of restrictions imposed on access to primary care clinics after the start of the Corona Virus Disease 2019 (COVID-19) pandemic, the recruitment of additional patients had to be conducted in clinics that at this timepoint, were in a severe lock-down situation.

After the intervention, a medical secretary, invited participants from the intervention group to participate in qualitative interviews. Inclusion criteria were participation in the 12-month follow-up and contributors from all sites. To balance the representation, both sexes and a variation in age were additional requirements.

## Randomization and allocation concealment

Participants from the cross-sectional survey that fulfilled the inclusion criteria for participation in the intervention study were contacted by telephone by a medical secretary, informed about the study, and invited to participate. An independent person using statistical software performed a block randomization stratified by study site using the ralloc-command in Stata with block sizes varying from 2 to 10. Allocations on the randomization list were numbered sequentially, participant number one was given allocation number one, et cetera. The authors did not have access to information that could identify participants during or after the data collection. After the external computer-based randomization and allocation, concealed from the investigators, the medical secretary informed them whether they were allocated to intervention or control group by another telephone call. Those who were allocated to the intervention group were scheduled for the 12-month follow-up at each of the four clinics, whereas people in the control group received care as usual. After randomization one patient in the intervention group withdrew consent and two patients in the control group.

Recruiting from regular consultations versus recruiting patients from clinics that only allowed patients with specific needs to attend face-to-face consultations because of the COVID-19 pandemic could have a profound implication for the analysis and interpretation of the data. We identified this as an extenuating circumstance [12] leading to an inclusion of a type of participants to the trial with more specific health challenges than described in the protocol. The treatment allocation was concealed for the nurse who enrolled participants in the study. Because of the nature of the intervention, we were unable to blind participants, data collectors, investigators, or clinicians.

## The intervention

Our intervention consisted of 1) an empowerment-based counselling program; 2) a standardized medical report:

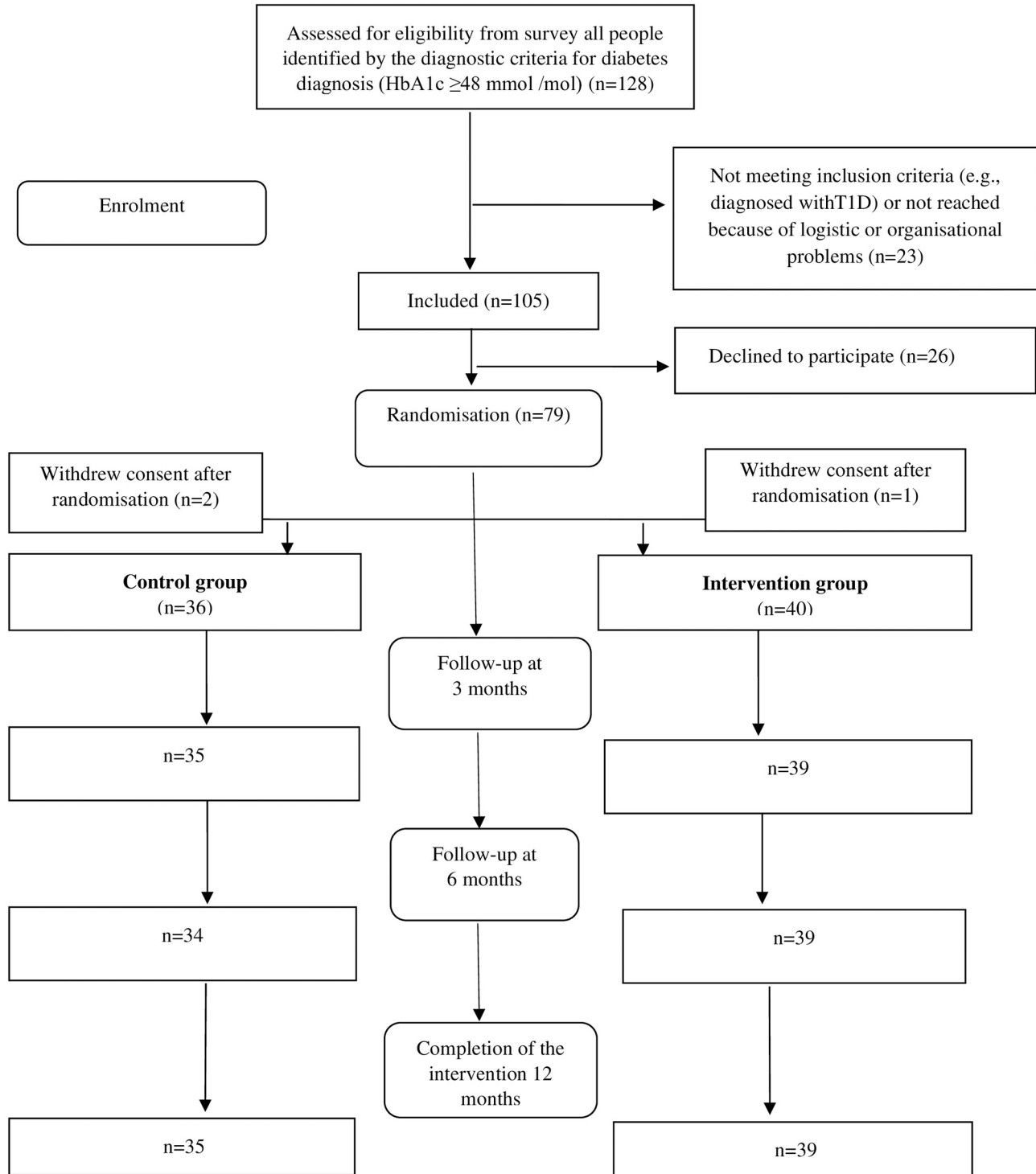

**Fig 1. Flow of eligible participants for inclusion in intention-to-treat analysis in the randomized trial.**

1. Empowerment-based counselling program. This was originally developed in Denmark with seven face-to-face consultations for people with type 1 diabetes [13, 14] using the Guided Self-Determination (GSD) approach, subsequently modified for use in primary care clinics among people with T2D for a Norwegian context [15]. In this current study we further adapted GSD into an interprofessional follow-up program consisting of a stepwise approach with more flexibility in the use of counselling tools such as the GSD reflection sheets during the consultations.

2. Standardized medical report. This was produced by the interprofessional team to facilitate the dialogue between nurses and physicians throughout the consultations and to ensure that current national clinical diabetes guidelines for the treatment of T2D and fidelity in applying the GSD principles were fulfilled during the consultations. Moreover, the GSD principles were included in the medical report to standardize its use across all participants during the consultations. Both nurses and physicians used the standardized medical report in the consultations.

The intervention was delivered as individual consultations, with one initial nurse consultation at baseline to get insights comprising the patients' thoughts and wishes for the cooperation with the healthcare professionals, followed by three face-to-face consultations over one year (three, six and 12 months). Nurses undertook an autonomous role in communication and counselling based on a structured agreement between nurses and physicians, beyond instrumental monitoring of blood pressure or other tasks and procedures delegated or ordered by the physicians. We provide further details of the 12-month program in S1 Fig and S1 Table in S2 File. In brief, besides using semi-structured reflection sheets, the nurses applied a set of communication skills (mirroring, active listening, and values clarification response) to support the participants' reflection on problem areas concerning their life and health with diabetes and encourage and activate individuals to better self-management.

## Training and collaboration

We trained healthcare professionals with 1) an introductory presentation for physicians and nurses of the GSD method and how to use it in this study; 2) two workshops, a GSD reading list and follow-up supervised exercises and individual observation sessions with a certificated GSD supervisor (BCHK) for nurses. To ensure fidelity in using the GSD method the nurses filled out a predefined reflection sheet after every fifth conversation focusing on how they used the communication skills (mirroring, active listening, and values clarification response). These reflections were discussed with the GSD supervisor (BCHK). Nurses had continuous access to the supervisor by e-mail and phone contact whenever they needed support and weekly phone meetings between the nurses and the GSD supervisor. We adapted the medical report to also include documentation on the nurses' use of GSD principles. Physicians developed and refined the standardized medical record, communicated with the GSD supervisor to get information about the nurses' training, and met regularly with the nurses to discuss psychological, organizational, and structural challenges.

## Control group

Participants in the control group received standard care consisting of individual consultations with the primary care physicians only and without the standardized medical report. Depending on physicians' assessments, follow-up was individualized with annual or more intensive schedules during the 12-month intervention period.

## Data collection

We collected data at baseline, three, six- and 12-months follow-up. We recorded the following sociodemographic data by a questionnaire; living situation (living with others or living alone), educational level (primary/middle school, secondary education, university/college ≤4 years, university/college >4 years), work situation (working full-time, working part-time, on benefits, retired, or other), and exercise regularity (never, less than weekly, once a week, 2–3 times a week, nearly every day) as an indication of physical activity level. As to clinical data, a medical secretary recorded the following data from medical records from December 1st, 2019, to June 15th, 2021: sex, age, glycosylated hemoglobin (HbA1c) (%, and mmol/mol), total cholesterol (mmol/l), High-Density Lipoprotein (HDL) (mmol/l) and Low-Density Lipoprotein (LDL) (mmol/l). The secretary also carried out the following measures: waist circumference (cm), weight (kg), and height (m) for all patients and echocardiogram (ECG) and monofilament in the intervention group. Measures described in current clinical guidelines for T2D (e.g. ECG, monofilament test) were only conducted in the control group if ordered by the physicians as these were the routines within standard care at all sites.

Our primary outcome was the Patient Activation Measure (PAM-13), a patient-reported outcome (PRO) measure completed by participants to assess individual knowledge, skills, and confidence integral to managing one's health and healthcare. The PAM-13 comprises 13 items on a 1–4-point Likert scale with responses ranging from "strongly agree" (4) to "strongly disagree" (1) or "not applicable" [16, 17]. The scale scores are summarized in a 0–100 activation scale, with higher scores representing higher activation. The questionnaire was translated into Norwegian by Steinsbekk and colleagues [18] and displayed satisfactory psychometric properties. Moljord et al [19] has also shown that the Norwegian version of PAM -13 has an acceptable factorial validity, a test-retest intraclass correlation coefficient of 0.76 and a statistically significant activation improvement ($p < 0.001$) supporting its use as a suitable tool to understanding activation in a clinical setting.

In addition to BMI, weight, waist circumference and HbA1c, our secondary outcomes were health status (EuroQoL(EQ)-5D-5L, EQ-VAS, WHO-Overall Health) and psychological well-being (WHO-5, WHO- Overall QOL, Problem Areas In Diabetes scale-5 (PAID-5)). The EQ-5D-5L measures health and functioning status within five domains: mobility, self-care, usual activities, pain/discomfort, and anxiety/depression [20–22]. Each dimension defines five levels (from no problem to extreme problems) converted into a single index value. In addition, it includes one item rating overall health on a visual analogue scale (EQ-5D-VAS) measuring the current health status from 0 (worst health imaginable) to 100 (best health imaginable). The scale was translated into the Norwegian language in accordance with EuroQol translation procedures including forward backwards translation, cognitive debriefing, and quality control [23] and further Norwegian norms have been provided by Garrett et al [24]. Additionally, we used one global item (WHO-Overall Health) from the WHO Quality of life-BREF scale to assess the participants' overall satisfaction with health [25]. The response options range from "very dissatisfied" (1) to "very satisfied" (5). The scale has been translated into Norwegian by Hanestad et al [26] confirming scaling qualities, discriminative power, and domain structure of the translated version. Further, the psychometric properties are further validated by Kalfoss et al [27]. Convergent and discriminant validity were evaluated as acceptable by examining their relationship between the four domains of WHO Quality of life-BREF and UWES-9, overall QoL and satisfaction with health using Pearson's product-moment correlation coefficient analysis. The four domains of WHO Quality of life-BREF were all positively correlated with work engagement, and with overall quality of life and satisfaction with health.

We used the five-item World Health Organization well-being index (WHO-5) to measure subjective psychological well-being during the previous two weeks. The scale is a 6-point Likert scale with a rating from 0 (none of the time) to 5 (all the time) [28, 29]. The raw score is transformed to a 0–100 scale, with lower scores indicating poorer well-being. Scores <50 are considered as reduced well-being, and scores <28 points indicate likely depression. According to the systematic review of the extensive body of literature on the WHO-5 [28], the scale has adequate validity both as a screening tool for depression and as an outcome measure in clinical trials in order to assess well-being over time. To assess the participants' overall satisfaction with life, we used the global quality of life item (WHO-Overall QOL) from the WHO Quality of Life-BREF scale [25]. The response option ranges from "very poor" (1) to "very good" (5), and the item reads: "How would you rate your quality of life?". As described above also the global quality of life item (WHO-Overall QOL) from the WHO Quality of Life-BREF scale display acceptable psychometric properties among Norwegians [27]. Finally, we used the PAID-5 to assess diabetes distress [30] The scale is a five-item short form of the original 20-item PAID scale measuring diabetes-related emotional distress in people with diabetes. Total scores range from 0–20 based on ratings on a five-point Likert scale with response options of 0–4, where higher scores imply greater diabetes distress and a score ≥8 indicates serious diabetes distress. The scale has been translated into Norwegian by Vislapuu et al [31]. The psychometric evaluation of the PAID-5 confirmed its postulated one-factor structure using confirmatory factor analysis (CFA). Convergent validity was demonstrated by statistically significant moderate correlations with other concept-related PROMs. The scale showed good internal consistency and a stable test-retest reliability confirming its usefulness for assessing diabetes-related emotional distress among patients with type 1 and type 2 diabetes in Norway.

After the intervention, we conducted individual interviews with participants. We conducted the interviews by telephone due to COVID-19 restrictions. The interview guide included topics such as experiences with participation in the intervention and whether the conversation with the nurses and physicians had led to any change in their ability to manage their health and health care, setting and achieving health goals (S2 Table in S2 File). The interviews lasted 30–40 minutes.

## Ethics

We obtained ethical approval for the study from the Regional Committee for Medical and Health Research Ethics South-East Norway (2019/28/REK south-east A) and the processing of personal data from the Norwegian Centre for Research Data (NSD:ID:821994). We conducted the project in accordance with the Helsinki Declaration. Further, we designed and reported the study in accordance with the CONSERVE-CONSORT Extension Guidelines for Reporting Trial Protocols and Completed Trials Modified Due to the COVID-19 Pandemic and Other Extenuating Circumstances [12] and Consolidated Criteria for Reporting Qualitative Research (COREQ) [32]. Further information can be obtained from ClinicalTrials.gov (ID: NCT04076384).

## Analysis

**Statistical analysis.**   We undertook descriptive analyses on demographic variables (sex, age, educational level, living situation and work situation) using mean value and standard deviation (SD) and confidence intervals (CI) for continuous variables and frequencies and percentages for categorical variables. We analyzed between-group differences at three, six- and 12-month follow-up for primary and secondary outcomes using linear mixed models with random intercept. The intervention effect was estimated as the interaction effect between group

and categorical time, using data in long format with up to four measurements per individual. By omitting the main effect of group from the model, we achieved an adjustment for baseline differences in the outcome [33]. We report the regression coefficient for the interaction term for each time point with 95% CI, which can be interpreted as the mean difference in the outcome between the intervention group and the control group at the given time point, adjusted for the baseline value of the outcome. Participants who dropped out during follow-up were included in analyses until the point of drop-out. In addition to assessing between-group differences, we undertook within-group analyses using linear mixed models with random intercept to test for change during follow-up separately for the intervention and control groups. The blood samples were analyzed using Afinion AS100/Alere™, a point-of care glycated hemoglobin measurement method validated for diagnosis and monitoring of diabetes by Stavelin et al [34]. All the hematologic analyses were the same at all four sites All statistical analyses were performed with SPSS® Statistics 28 (IBM Corp., Armonk, NY, USA) and STATA SE 16.0 and MP 17.0 (StataCorp LLC, College Station, Texas, USA).

**Qualitative analysis.** We used thematic analysis [35] to analyze qualitative data from the individual interviews. Thematic analysis is a framework and a formalized method within the psychological qualitative analysis field and healthcare research [36]. We transcribed the interviews verbatim. Two authors (MG, BCHK) read the transcribed data several times to get familiar with the data. We coded the data separately to identify patterns within and across the interviews. During the coding process we sought for ideas and patterns, and we wrote them down to use them in our ongoing discussion. We aimed at giving full attention to every single item in the data and identified interesting aspects that were the basis of some repeated patterns across our data set. After identifying potential overarching themes, we invited the full group of authors to comment on the themes and interpretation of the qualitative data. Finally, we refined the themes and revised the report. In the analysis process, we recognized the authors' various competencies. The first author who had a more peripheral role in the project, worked primarily in a university setting. This concern was compensated by the other co-authors' (KIB, BCHK, IH, AH), who had more extensive clinical expertise obtained from years of working as an endocrinologist or diabetes specialist nurses and some with first-hand experience living with diabetes.

## Results

Initially, we invited 105 people with T2D to take part in the study. We did not have success with the strategy to include additional patients from the clinics that at this timepoint were in a lock-down situation because of the COVID-19 restrictions. Thus, the anticipated effects of implementing mitigating strategies leading to an inclusion of a type of participants to the trial with more specific health challenges than described in the protocol did not appear. Participants had a mean age of 63.5 years (SD 10.54), 51.3% (n = 39) were men, 24.3% (n = 18) were living alone, 31.6% (n = 24) worked full-time and 44.7% (n = 34) were retired, with minor differences between the groups (Table 1). We invited nine patients from all four sites to participate in the qualitative interviews and eight consented: four male and four female, mean age 66.8 years (range 55–78 years).

### Quantitative data

At baseline, the mean PAM-13 score (SD) was 68.3 (12.8) and 76.3 (13.4) in the intervention and control groups, respectively (Table 2). We found no significant between-group differences at either time point. The mean between-group differences with 95% CI at three, six and 12 months after adjustment for baseline PAM scores were, respectively, -2.5 [-8.1, 3.1], 1.87 [-3.6,

**Table 1. Sociodemographic characteristics and health-related behaviors.**

| | Total | Intervention | Control |
|---|---|---|---|
| | N = 76 | n = 40 | n = 36 |
| Sex, n (%) | | | |
| Male | 39 (51.3) | 21 (52.5) | 18 (50.0) |
| Female | 37 (48.7) | 19 (47.5) | 18 (50.0) |
| Age, mean (SD) | 63.5 (10.54) | 62.9 (10.85) | 64.2 (10.28) |
| Living situation, n (%) | | | |
| Live with others | 56 (75.7) | 29 (76.3) | 27 (75.0) |
| Live alone | 18 (24.3) | 9 (23.7) | 9 (25.0) |
| Educational level, n (%) | | | |
| Primary/middle school | 24 (32.0) | 14 (35.9) | 10 (27.8) |
| Secondary education | 36 (48.0) | 18 (46.9) | 18 (50.0) |
| University/college ≤4 years | 9 (12.0) | 5 (12.8) | 4 (11.1) |
| University/college > 4 years | 6 (8.0) | 2 (5.1) | 4 (11.1) |
| Work situation[1], n (%) | | | |
| Full-time work | 24 (31.6) | 15 (37.5) | 9 (25.0) |
| Part-time work | 10 (13.2) | 5 (12.5) | 5 (13.9) |
| On benefits | 5 (6.6) | 3 (7.5) | 2 (5.6) |
| Retired | 34 (44.7) | 16 (40.0) | 18 (50.0) |
| Other | 3 (3.9) | 1 (2.5) | 2 (5.6) |
| Total cholesterol (mmol/l), mean (SD) | 5.28 (576) | 4.64 (2.02) | 5.98 (8.05) |
| HDL (mmol/l), mean (SD) | 1.23 (0.30) | 1.16 (0.22) | 1.30 (0.35) |
| LDL (mmol/l), mean (SD) | 2.40 (1.03) | 2.58 (1.12) | 2.21 (0.89) |
| ECG[2] performed | 26 (34.2) | 23 (57.5) | 3 (8.3) |
| Monofilament test | 41 (54.0) | 30 (75.0) | 5 (13.9) |
| BMI[3] (kg/m2, categories), n (%) | | | |
| <25 kg/m$^2$ | 10 (14.3) | 6 (15.4) | 4 (12.9) |
| 25–30 kg/m$^2$ | 32 (45.7) | 20 (51.3) | 12 (38.7) |
| ≥30 kg/m$^2$ | 28 (40.0) | 13 (33.3) | 15 (48.4) |
| Exercise regularity, n (%) | | | |
| Never | 3 (4.1) | 2 (5.4) | 1 (2.8) |
| Less than weekly | 10 (13.7) | 3 (8.1) | 7 (19.4) |
| Once a week | 10 (13.7) | 5 (13.5) | 5 (13.9) |
| 2–3 times a week | 23 (31.5) | 13 (35.1) | 10 (27.8) |
| Nearly every day | 27 (37.0) | 14 (37.8) | 13 (36.1) |

[1]"Other" includes leave of absence, home staying (without pay), under education, unemployed and other.

[2]ECG; echocardiogram,

[3]BMI; Body Mass Index.

Number of missing values for each variable in brackets: Age (1), Living situation (2), Education (1), Total cholesterol (1), HDL cholesterol (2), LDL cholesterol (2), BMI (6), Exercise (3).

7.4] and -4.2 [-9.6, 1.3]. We found no significant within-group changes in either group (Fig 2 and S3 Table in S2 File).

We found a significant between-group difference in HbA1c after 12 months with (B [95% CI] = -8.6 [-17.1, -0.1] mmol/l; p = 0.045) indicating 8.6 mmol/mol lower mean HbA1c compared to the control group after adjustment for baseline HbA1c (Table 2). We found no other significant between-group differences for any other secondary outcomes. When analyzing

**Table 2. Effect of the GSD intervention among people with type 2 diabetes in primary care (n = 76).**

| | Baseline values, mean (SD) | | Intervention effect 3 months | | | Intervention effect 6 months | | | Intervention effect 12 months | | |
|---|---|---|---|---|---|---|---|---|---|---|---|
| | Control n = 36 | Intervention n = 40 | n | B* (95% CI) | p-value | n | B* (95% CI) | p-value | n | B* (95% CI) | p-value |
| Primary outcome | | | | | | | | | | | |
| PAM-13 score | 68.3 (12.8) | 76.3 (13.4) | 67 | -2.5 (-8.1, 3.1) | 0.38 | 71 | 1.87 (-3.6, 7.4) | 0.51 | 72 | -4.2 (-9.6, 1.3) | 0.14 |
| Secondary outcomes | | | | | | | | | | | |
| BMI (kg/m$^2$) | 30.3 (4.5) | 29.1 (4.6) | 70 | -0.4 (-1.1, 0.3) | 0.25 | 71 | 0.0 (-0.7, 0.7) | 0.99 | 71 | -0.2 (-0.9, 0.5) | 0.59 |
| Weight (kg) | 92.4 (17.7) | 87.2 (15.9) | 72 | -0.1 (-2.7, 2.4) | 0.91 | 71 | -0.5 (-3.1, 2.0) | 0.68 | 72 | -1.6 (-4.2, 0.9) | 0.21 |
| Waist circumference (cm) | 107.9 (12.6) | 106.2 (11.5) | 64 | -3.0 (-6.9, 1.0) | 0.14 | 70 | -1.2 (-5.0, 2.6) | 0.53 | 68 | -3.2 (-7.1, 0.6) | 0.10 |
| HbA1c, mmol/mol | 51.9 (7.8) | 52.4 (11.1) | 73 | -1.9 (-10.4, 6.7) | 0.67 | 72 | -2.9 (-11.5, 5.7) | 0.51 | 74 | -8.6 (-17.1, -0.1) | 0.048 |
| WHO-5 | 69.4 (14.8) | 70.6 (16.5) | 69 | -2.7 (-9.1, 3.7) | 0.41 | 69 | -3.4 (-9.8, 3.1) | 0.31 | 68 | 3.4 (-3.1, 9.8) | 0.31 |
| WHO-Overall QOL | 3.9 (0.8) | 4.0 (0.7) | 69 | -0.2 (-0.5, 0.1) | 0.23 | 70 | -0.0 (-0.3, 0.3) | 0.84 | 72 | -0.0 (-0.3, 0.3) | 0.92 |
| PAID5 | 4.6 (4.2) | 3.9 (3.8) | 61 | -0.1 (-1.6, 1.4) | 0.85 | 68 | 0.2 (-1.3, 1.6) | 0.84 | 71 | 0.3 (-1.1, 1.7) | 0.66 |
| EQ-5D-5L | 0.9 (0.1) | 0.8 (0.2) | 61 | 0.0 (-0.0, 0.1) | 0.42 | 70 | 0.1 (-0.0, 0.1) | 0.051 | 72 | 0.0 (-0.0, 0.1) | 0.27 |
| EQ-5D-VAS | 70.6 (20.0) | 72.3 (19.9) | 62 | -0.9 (-7.9, 6.2) | 0.81 | 68 | -3.1 (-10.0, 3.7) | 0.37 | 70 | -2.1 (-8.9, 4.6) | 0.54 |
| WHO-Overall Health | 3.3 (0.8) | 3.4 (0.8) | 68 | -0.1 (-0.4, 0.2) | 0.71 | 70 | -0.0 (-0.3. 0.3) | 0.86 | 72 | -0.0 (-0.3, 0.3) | 0.95 |

*Regression coefficient for interaction term between categorical time and group allocation from linear mixed model with random intercept for individual. Adjusted for baseline value of the outcome by omitting main effect of group from the model.

within-group changes, we found a significant within-group weight reduction of -1.8 kg in the intervention group at 12 months. In comparison, the control group's within-group weight change of -0.5 kg at the same time point, was non-significant (S3 Table in S2 File). Accordingly, we found a significant within-group change of -3.9 cm for waist circumference in the intervention group at 12 months, whereas the within-group change in the control group (-1.0 cm) was non-significant. The mean scores for the health measures (EQ-5D-5L, EQ-5D-VAS and WHO-Overall Health) (Table 2) were relatively high at baseline, indicating that most participants did not report a high disease burden. None of the participants reported WHO-5 scores suggesting likely depression (<28), and only 8% scored lower than 50 (reduced well-being). Accordingly, none of the participants reported a PAID-5 score indicating serious diabetes distress (score ≥8). The means and 95% CI for the secondary PRO data at baseline, three, six and 12 months are displayed in Fig 3.

Apart from a significant decrease in WHO-5-scores in the intervention group at three months (B [95% CI] = -5.7 [-11.2, -0.2], p = 0.04) and six months (B [95%CI] = -6.1 [-11.6, -0.7], p = 0.03), there were no significant changes over time.

Regarding the examinations from the standardized medical report, 30 intervention group participants and two control group participants were tested with a monofilament test at the 12-month follow-up. Furthermore, 23 in the intervention group and two control group participants received ECG testing.

## Qualitative data

Our analysis identified two themes: *"Other thoughts and perspectives came up front"* and *"Although activated, I must find my own way and pace"*.

**Other thoughts and perspectives came up front.** The participants experienced that the GSD counselling approach gave rise to conversations during the follow-up that opened for new ideas and reflections. As other insights and understanding came up front, participants had the opportunity to elaborate on what they felt was important for them in their life situation

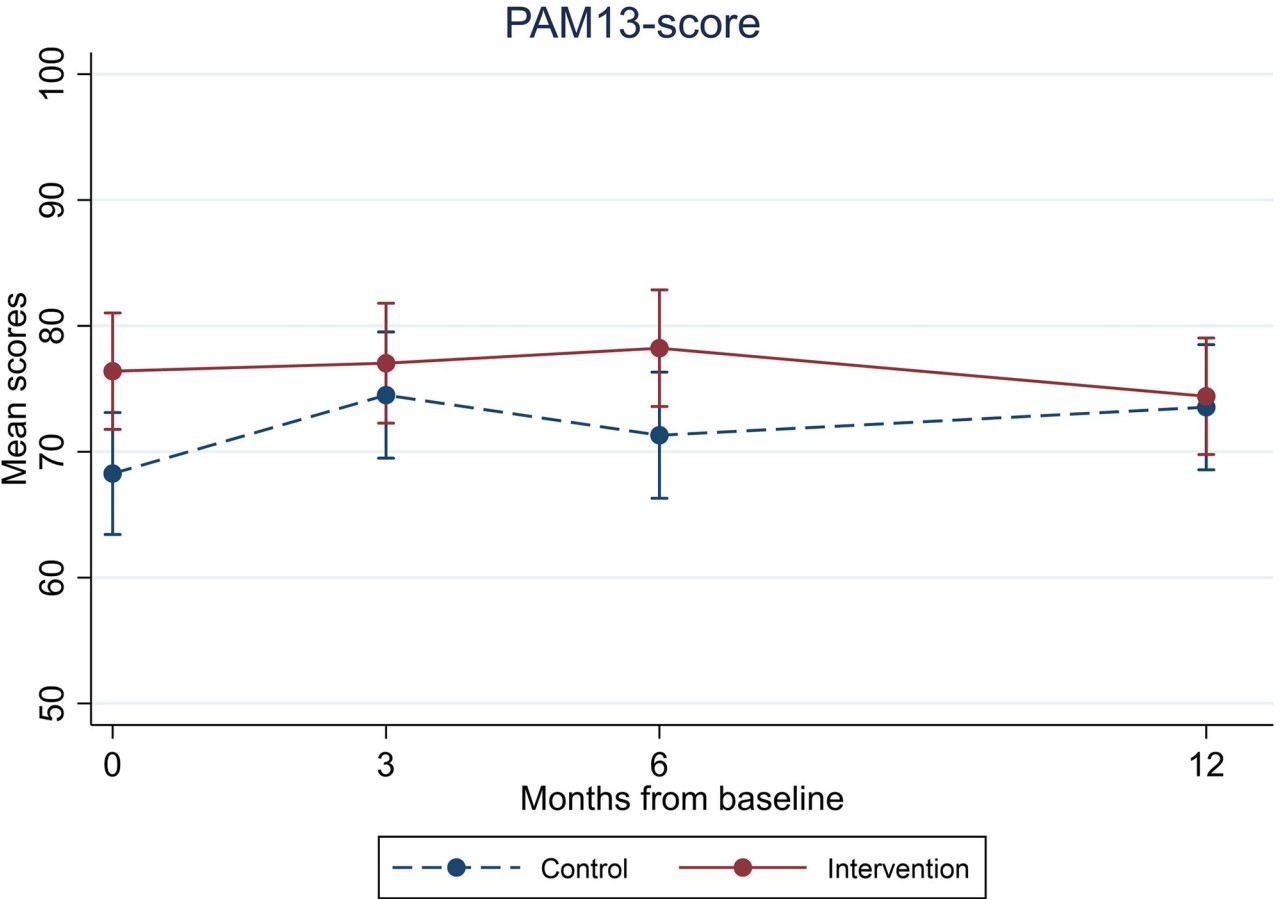

**Fig 2. Unadjusted mean PAM-13 score with 95% confidence intervals in the intervention group and the control group at baseline and during follow-up (color should be used for this figure in print).**

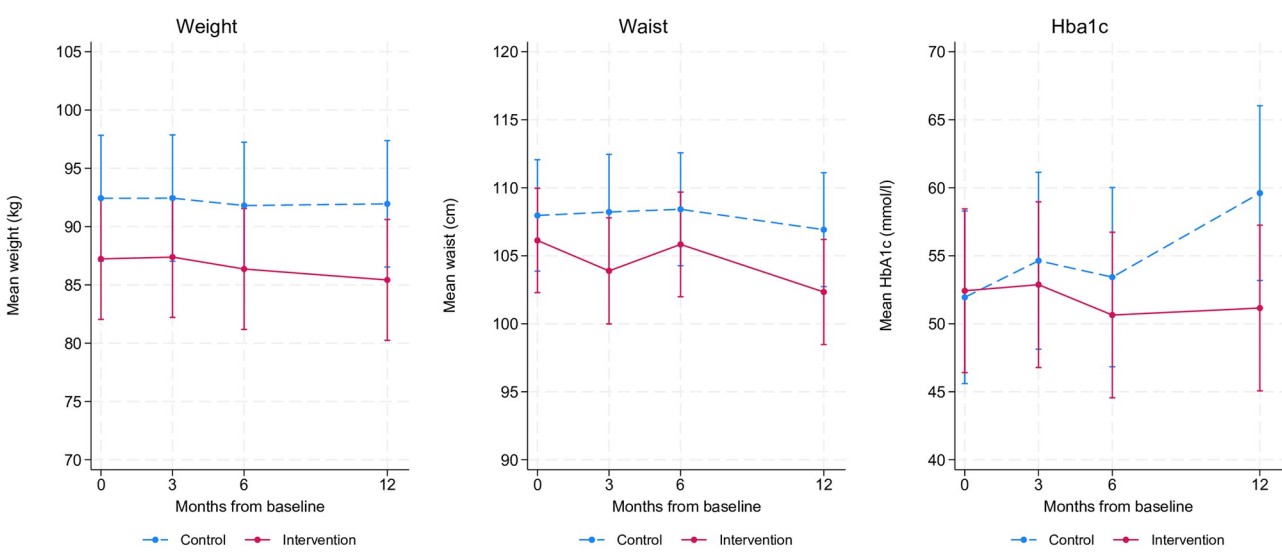

**Fig 3. Unadjusted means and 95% CI of the secondary outcome measures weight, waist circumference and HAa1c in the control group and intervention group at baseline and during follow-up (color should be used for this figure in print).**

at that moment. In this way, dialogues on important matters in life that also affected their diabetes, came about. Essentially, the participants had access to the expertise of the physicians. At the same time, the nurse consultations meant that they could come forward with matters they had in mind, but not necessarily expressed during regular consultations with health care professionals. One participant said:

> . . .. *the nurse kind of follows up all the way, and she knows if there's something extra that will take place. . .. you are constantly getting new questions about different things. . .so I feel very safe*
>
> *(Informant 1).*

One essential aspect of the follow-up was that the participants perceived that they could express their frustration or feelings about living with diabetes more in full. Being open to such feelings might help an individual to realize what hinders a more constructive behavioral health change process. Some patients expressed that although they managed to live well with their diabetes, they felt that talking with the nurse about the hassle of having diabetes was helpful. That the GSD consultations were conducted in dialogue with someone who knew the patients in persona and thus could challenge them beyond previous conversations in regular consultations at the clinics seemed essential. Also, the dialogue gave a kind of validation of the patients' competence and thoughts about living with diabetes. For example, one patient stated:

> *Nothing new came to light. I have pretty good control over this myself. But it has been nice that others have agreed with my experiences*
>
> *(Informant 7).*

**Although activated, I must find my own way and pace.**   The participants expressed that the consultations opened for a greater awareness of health challenges, but it was still up to each person to act. They expressed that the interprofessional follow-up facilitated for more reflection and an increase in their motivation to undertake changes in their life. One patient said:

> *I think the nurse is very good to talk to . . . I could talk about what matters for me. I became motivated to think more about what I was eating.. ..*
>
> *(Informant 8).*

However, the patients experienced that they had to find their own way to manage their life situation as best as possible. It was not the lack of knowledge that hindered them in reaching their own goals, but rather a lack of inner motivation. They expressed that although they had started to reflect on their health behavior, it is not a quick fix. They emphasized that it had to be done at their own pace. One said that he intended to act, but:

> . . .. *because you can't flip the shilling as they say, it will always take some time,. . . maybe I've become more aware of healthier food choices then. Although I still eat candy and all that anyway, but maybe not the same amount. . .*
>
> *(Informant 2).*

Although some participants felt that they had started a process to improve health behaviors, such as eating bread that is a bit coarser and different things like that, others pointed out that also circumstances beyond their control came about:

*I was good at exercising. I was out walking. Corona made this worse when the fitness centers were closed. The weather has prevented me from going out as well. . .. there has been less motivation. I feel that some external circumstances have an influence*

*(Informant 6).*

## Discussion

We found no effect on the patient activation measure in the population investigated. Nevertheless, patients emphasized that the intervention brought about a greater understanding of health challenges. They had started a process to improve health behaviors because the intervention had opened their eyes to new reflections and a greater awareness of their ways of living. Thus, the differences we observed in HbA1c, body weight and waist circumference in favor of the intervention group were further illuminated by the qualitative data. The findings made clear that the participants needed to find their own way and time to take on actions.

The results indicate that the participants in this rather healthy T2D population perceived that they could manage their daily life with the condition. Biomarkers as well as findings from the interviews showed that the patients did not express that they had severe symptoms or high burden of managing their health care. Thus, it is not unexpectedly that the PRO scales demonstrated a profound ceiling effect. Overall, participants reported relatively high patient activation scores at baseline and follow-up. The results indicate that the participants perceived that they could handle information to help them to better self-manage their health. This finding indicates that using primarily generic instruments in our population might not be sensitive enough to capture change in perception of health and well-being. Previous studies suggesting that self-report scales lack discriminative ability for more healthy populations indicates that they must be used with caution among people with diabetes displaying scores close to the normal population [22, 37]. It is shown that although patient-reported scales such as the EQ-5D, have the ability to distinguish between different complications and levels of severity, ceiling effects are more profound among populations who report no problems on any dimension and hereby rate themselves as in full health [22].

However, some participants expressed frustration about having diabetes in the qualitative part of the study. To be allowed to share such thoughts and anger in consultations might be essential for self-care activities. Undoubtedly, self-management support that helps people to accept the disease is important to enable patients' confidence in making treatment choices and taking care of their diabetes [4]. One might argue that the intervention activated thoughts of taking steps towards healthier choices in life, leading to better self-management. Greater attention to weight management as part of a holistic approach to diabetes self-management is more emphasized now than earlier [3]. Studies have demonstrated that weight reduction of 10–15% can have a disease-modifying effect and lead to remission of T2D with normal blood glucose levels without any pharmacological therapy [3, 38]. Thus, the reduction in weight and waist circumference in the intervention group is helpful. To illuminate this matter further a longer intervention period of for instance 2–5 years might have said more. Still, concerns have been made from the perspective of primary care practitioners on the mismatch between resources needed to facilitate access to effective obesity diagnosis and treatments and the organizational structure and capacity of practices [39].

Although small, the mean difference in glycemic control in favor of the intervention group is a positive finding. There is significant evidence that smaller HbA1c improvements are likely to reduce the risk of diabetes complications [2, 40, 41]. A meta-review of quantitative systematic reviews on self-management support interventions for people with T2D has shown that the majority of the reviews indicate HbA1c improvements between 0.2% and 0.6% (2.2 to 6.5 mmol/mol) at six months follow-up [5]. Further, comparative effectiveness meta-analyses suggest that each new class of noninsulin agents added to initial therapy with metformin generally lowers HbA1c by approximately 0.7–1.0% [42]. Such results are potentially important as the UKPDS study demonstrated that each 1% (11 mmol/mol) reduction in HbA1c was associated with a significantly reduced risk of micro- and macrovascular complications, as well as a decrease in risk of diabetes-related deaths [2]. Thus, evidence that self-management support is essential to reduce the risk of all-cause mortality is important [43].

In terms of meeting clinical diabetes guidelines, it has previously been reported that there are major gaps in the performance of recommended screening procedures to detect microvascular complications in Norwegian primary care practices [44]. In this study, it was a positive finding that the nurses in the intervention arm followed the guidelines regarding annual recommended examinations (monofilament, ECG) to a greater extent than in the control arm. Strengthening the follow-up by a more integrated interprofessional collaboration has previously been shown to positively impact patient outcomes in primary care [8].

Our findings show that the intervention could have a meaningful impact when starting a change process. However, achieving and sustaining long-term changes in health behavior requires continuous education, behavioral and psychosocial support [45]. One might argue that the intensity and length of this rather low dose interprofessional intervention might have been too weak and too short. In individual interviews, patients emphasized that it was difficult to stay caught up and maintain some motivation because of so few consultations and the relative long follow-up intervals. Previous studies show that the intensity of an intervention influences its effectiveness [5]. It has been demonstrated that the best outcomes from self-management support programs are achieved through programs that have a longer duration (more than 10 hours), are structured, and have a defined theory base [4, 46]. In other patient activation intervention studies, it has been shown that longer duration of follow-up is associated with larger HbA1c improvements [47]. Nonetheless, to align with the responsibilities and workload challenges in primary care, we adapted this intervention into a minimized follow-up format for use in these services. As such, we intended to evaluate a low dose intervention program. Unfortunately, this dose was insufficient to promote substantial change in health behavior. In addition, capacity problems due to altered organizational routines and a greater workload imposed by the COVID-19 pandemic hindered further recruitment from fulfilling the power calculation estimate of the number of participants needed in this trial. The overall implications for this intervention were that the eligibility criteria and recruitment strategies may not have been sufficient to include enough patients to show benefit.

There are several limitations to this study. Our sample size was small, primarily due to the recruitment challenges. The pandemic also made it difficult for the participants to engage in physical activity as fitness centers were closed, and people were not allowed to meet face-to-face. Thus, both physical and social incentives to keep on track faded out. It might be a limitation that we did not collect data from qualitative interviews with participants in the control group as such data might have shed light on everyday life during the pandemic also for this group. Furthermore, we primarily used generic PROs, which demonstrated methodological problems due to ceiling effects and lack of sensitivity. Likewise, mean HbA1c and findings from the interviews showed that the study sample consisted of adults with relatively well-regulated T2D with less potential to change health outcomes. Therefore, these outcome measures

had limited potential for improvement. Finally, patients invited to the study were recruited from scheduled consultations which could indicate that they did not have explicit health problems and little motivation to engage in healthier behaviors and undertake new habits and activities. Although the inclusion of all persons in the waiting room area indicates that our results are representative for the general population attending a GP appointment, we did not have more detailed information on additional diagnosis or underlying conditions for these patients. Although underpowered to detect differences in scale scores, the RCT design was a strength. Furthermore, few participants dropped out of the study from either group. Only, two, three and two participants did not meet at the 3-, 6- and 12-months visits, respectively (Fig 1).

## Conclusions

The results from this study did not document any significant effect on self-reported questionnaire outcomes in the population investigated. However, the significant ceiling effects, represented by the relatively high scores at baseline, indicated that most participants did not have a large disease burden or that the instruments used might not capture what matters the most for these patients. The qualitative data showed that the intervention opened patients' eyes to reflection, entailing a greater awareness of health challenges. The dialogues on important matters in their life situation illuminated the potential beneficial effects on HbA1c and body weight that was identified. We suggest testing the intervention in a group of patients with larger disease burden and a more expressed motivation for change.

## Supporting information

**S1 Checklist.**
(DOCX)

**S1 Fig. Overview of the GSD program and follow-up consultations over 12 months.**
(TIF)

**S1 File.**
(DOCX)

**S2 File.**
(DOCX)

## Acknowledgments

We thank all four primary care clinics who participated in the intervention. We are especially grateful to the nurses for their invaluable effort applying GSD in the consultations.

## Author Contributions

**Conceptualization:** Marit Graue, Jannicke Igland, Anne Haugstvedt, Vibeke Zoffmann, David A. Richards, Beate-Christin Hope Kolltveit.

**Data curation:** Marit Graue, Jannicke Igland, Anne Haugstvedt, Ingvild Hernar, Kåre I. Birkeland, David A. Richards, Beate-Christin Hope Kolltveit.

**Formal analysis:** Marit Graue, Jannicke Igland, Anne Haugstvedt, Ingvild Hernar, Kåre I. Birkeland, David A. Richards, Beate-Christin Hope Kolltveit.

**Funding acquisition:** Marit Graue, Beate-Christin Hope Kolltveit.

**Investigation:** Marit Graue, Jannicke Igland, Anne Haugstvedt, Ingvild Hernar, Kåre I. Birkeland, Vibeke Zoffmann, David A. Richards, Beate-Christin Hope Kolltveit.

**Methodology:** Marit Graue, Anne Haugstvedt, Vibeke Zoffmann, David A. Richards, Beate-Christin Hope Kolltveit.

**Project administration:** Marit Graue, Beate-Christin Hope Kolltveit.

**Resources:** Marit Graue.

**Supervision:** Vibeke Zoffmann.

**Writing – original draft:** Marit Graue, Jannicke Igland, Anne Haugstvedt, Ingvild Hernar, Kåre I. Birkeland, Vibeke Zoffmann, David A. Richards, Beate-Christin Hope Kolltveit.

**Writing – review & editing:** Marit Graue, Jannicke Igland, Anne Haugstvedt, Ingvild Hernar, Kåre I. Birkeland, Vibeke Zoffmann, David A. Richards, Beate-Christin Hope Kolltveit.

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
