## [Decision Letter · Decision Letter 0]

21 Jun 2023

PONE-D-23-11188Evaluation of an interprofessional follow-up intervention among people with type 2 diabetes in primary care – a randomized controlled trial with embedded qualitative interviews.PLOS ONE

Dear Dr. Graue,

Thank you for submitting your manuscript to PLOS ONE. After careful consideration, we feel that it has merit but does not fully meet PLOS ONE’s publication criteria as it currently stands. Therefore, we invite you to submit a revised version of the manuscript that addresses the points raised during the review process.

We look forward to receiving your revised manuscript.

Kind regards,

Jennifer Annette Campbell, PhD, MPH

Academic Editor

PLOS ONE

“I have read the journal's policy and the authors of this manuscript have the following competing interests: KIB has received research support from Astra Zeneca, Bayer, Boehringer Ingelheim, Lilly, MSD, Novo Nordisk, Roche, Sanofi and Sysmex Norway. The other authors have no relevant competing interest to disclose.”

Reviewers' comments:

Reviewer's Responses to Questions

**Comments to the Author**

1. Is the manuscript technically sound, and do the data support the conclusions?

Reviewer #1: Yes

Reviewer #2: Partly

Reviewer #3: Yes

2. Has the statistical analysis been performed appropriately and rigorously? 

Reviewer #1: Yes

Reviewer #2: No

Reviewer #3: No

3. Have the authors made all data underlying the findings in their manuscript fully available?

Reviewer #1: Yes

Reviewer #2: Yes

Reviewer #3: No

4. Is the manuscript presented in an intelligible fashion and written in standard English?

Reviewer #1: Yes

Reviewer #2: Yes

Reviewer #3: Yes

5. Review Comments to the Author

Reviewer #1: This is a good study, but I have suggestions for the authors.

In the inclusion criteria, it is unclear whether only patients with T2DM or other underlying conditions and T2DM are included.

The assignment was not explained. Please add it.

Please also indicate secondary outcomes in the methods section.

Please indicate whether the same automated hematology analyzer did all hematologic analyses in four clinics.

Please indicate that you use the Norway version of mentioned scales or the original one. Also, if you used the Norway version, please indicate the reliability and validity of each.

Reviewer #2: The study aimed to evaluate the effects of an empowerment-based interprofessional follow-up intervention in people with T2D in primary care on patient-reported outcomes, biomarkers and weight, and to explore the experiences of patients attending the intervention.

The manuscript could be improved based on the following comments.

Line 87 – 89, the sentence is unclear and requires revision such on the enrollment period, intervention period, follow-up period, and end of study.

Line 91-97, more detail information on the inclusion and exclusion criteria is to be provided.

Line 101-103, the software which was used in the sample size calculation is to be stated.

Line 110 – what about the control group?

The effect size index could be presented.

Line 239, analysis is to be written as Statistical Analysis.

Line 251-252, the sentence unclear and requires revision.

Any missing data is to be highlighted including the percentage, pattern etc.

The percentage figures stated in the text is to be at least one decimal point.

Table 1, n for Total could be replaced with N.There are discrepancies in the sample size. If there are missing data or not presented, it is to be presented/highlighted.

Table 2, statistical test and B is to be denoted in table footnote. If asterisk is to be used in the footnote, the asterisk symbol is to be indicated in the table e.g. B with superscript asterisk. A description of n at 3 months, 6 months and 12 months is to be stated in the table footnote and any missing data is to be denoted.

Line 310, it is not SI Fig 1 but supplementary Table 3.

Need to revise the cited Table in Line 296 and Line 314.

The supplementary tables numbering in the attachments need to be properly numbered and aligned to the cited table in the text.

For the qualitative interview, it would be good to cite the participants who gave the same opinions/statements so that we know all the 9 participants were involved in the discussion.

Figure 1 and Line 469, the loss of follow-up and reason is to be indicated/stated in the figure.

Figure 3 is blur, small font and hard to visualize.

CONSERVE Checklists incomplete.

For CONSORT Checklist, some page numbers are not tally with the manuscript page numbers.

Reviewer #3: 1) The authors stated in lines 96-97: “79 consented to participate: 40 people in the intervention group and 36 in the control group.” What happened to the other three individuals who were consented?

2) How did the authors check for fidelity? Were the counseling sessions recorded? How do you confirm the counseling sessions were standardized across all participants?

3) Please explain how you chose the covariates for adjustment. Did you use variable selection in your analysis?

4) What time point HbA1c was used for your analysis? how did you handle the scenario if there were multiple HbA1c records within this time period?

5) The authors stated, “There is significant evidence that smaller HbA1c improvements are likely to reduce the risk of diabetes complications.” Can the authors provide more citation with regards to this statement?

6) This study with a sample size of 76 is very underpowered and the effects seen are marginally significant. How clinically significant are these results?

7) The authors stated in lines 284-285: “We invited nine patients from all four sites to participate in the qualitative interviews and eight consented; four male and four female, mean age 66.8 years (range 55-78 years).” What was the criteria used to invite participants for the interviews? Is it a convenient sampling? What are the associated demerits or limitations?

6. PLOS authors have the option to publish the peer review history of their article (what does this mean?). If published, this will include your full peer review and any attached files.

Reviewer #1: **Yes: **Mahdi Abounoori

Reviewer #2: No

Reviewer #3: No

---

## [Author Response · Author response to Decision Letter 0]

2 Aug 2023

Thank you for reviewing our manuscript for publication in PLOS ONE, and for constructive comments from the academic editor and reviewers. We have now answered all the questions and revised the manuscript accordingly. Below are the comments from reviewers (in italic) followed by our responses (Re). All page and line numbers refer to the uploaded pdf file labeled 'Revised Manuscript with Track Changes'.

Journal requirements

Re: Thank you for providing information upon the PLOS ONE style templates. We have revised the manuscript accordingly. 

2) Thank you for stating the following in the Competing Interests section: “I have read the journal's policy and the authors of this manuscript have the following competing interests: KIB has received research support from Astra Zeneca, Bayer, Boehringer Ingelheim, Lilly, MSD, Novo Nordisk, Roche, Sanofi and Sysmex Norway, AH has received honoraria for speaking at meetings from Novo Nordisk and Abbot. IH has received honoraria for speaking at meetings from Novo Nordisk. The other authors have no relevant competing interest to disclose.” Please confirm that this does not alter your adherence to all PLOS ONE policies on sharing data and materials, by including the following statement: "This does not alter our adherence to PLOS ONE policies on sharing data and materials.” (as detailed online in our guide for authors http://journals.plos.org/plosone/s/competing-interests). If there are restrictions on sharing of data and/or materials, please state these. Please note that we cannot proceed with consideration of your article until this information has been declared. Please include your updated Competing Interests statement in your cover letter; we will change the online submission form on your behalf.

Re: We have stated this information in the cover letter as requested. 

3) We note that you have indicated that data from this study are available upon request. PLOS only allows data to be available upon request if there are legal or ethical restrictions on sharing data publicly. For information on unacceptable data access restrictions, please see http://journals.plos.org/plosone/s/data-availability#loc-unacceptable-data-access-restrictions. In your revised cover letter, please address the following prompts: If there are ethical or legal restrictions on sharing a de-identified data set, please explain them in detail (e.g., data contain potentially identifying or sensitive patient information) and who has imposed them (e.g., an ethics committee). Please also provide contact information for a data access committee, ethics committee, or other institutional body to which data requests may be sent.

Re: We have addressed the following in our revised cover letter: Due to personal data protection legislation and legal restrictions related to confidentiality, the data cannot be deposited online as the study participants have not explicitly been informed about, nor approved data sharing when the data were gathered in 2019-2021 (see approval from The Regional Committee for Medical and Health Research Ethics South-East Norway (2019/28/REK south-east A, and the processing of personal data from the Norwegian Centre for Research Data (NSD:ID:821994)). Thus, the data in the current study is not publicly available due to ethical regulatory conditions of data usage. However, the ethics committee allows us to include collaborators to participate if the research questions align with the research question and purpose of the study that the participants have been informed about. Thus, collaborators may contact the first author to put such applications forward to the Regional Committee for Medical and Health Research Ethics South-East Norway for approval. 

Reviewer #1

1) This is a good study, but I have suggestions for the authors. In the inclusion criteria, it is unclear whether only patients with T2DM or other underlying conditions and T2DM are included.

Re: We included all people with T2D identified by the diagnostic criteria for diabetes diagnosis (HbA1c ≥48 mmol /mol) who had participated in a cross-sectional survey conducted in the waiting area among adults aged 20-80 years scheduled for regular consultations at the four clinics from May 2019 to December 2019. The inclusion of all persons in the waiting room area indicates that our results are representative for the general population attending a GP appointment. It is unfortunate that we did not have more detailed information on additional diagnosis or underlying conditions for these patients. We have added this to the limitation section (page 27, line 527-530).

2) The assignment was not explained. Please add it.

Re: Thank you, we have revised the text to make this clearer. We have explained randomization and allocation concealment. In addition, we have included the following text on how patients were assigned to each group (page 6). “Participants from the cross-sectional survey that fulfilled the inclusion criteria for participation in the intervention study were contacted by telephone by a medical secretary, informed about the study, and invited to participate.” (lines 123-125), and “After the external computer-based randomization and allocation, concealed from the investigators, the medical secretary informed them whether they were allocated to intervention or control group by another telephone call. Those who were allocated to the intervention group were scheduled for the 12-month follow-up at each of the four clinics, whereas people in the control group received care as usual. After randomization one patient in the intervention group withdrew consent and two patients in the control group.” (lines 130-135). 

3) Please also indicate secondary outcomes in the methods section.

Re: Thank you for this comment. We have added information on this matter in the Methods section (page 10, line 222).

4) Please indicate whether the same automated hematology analyzer did all hematologic analyses in four clinics.

Re: Thank you for this comment. We have included the following text at page 13, lines 303-306. “The blood samples were analyzed using Afinion AS100/AlereTM, a point-of care glycated hemoglobin measurement method validated for diagnosis and monitoring of diabetes by Stavelin et al (Stavelin et al 2020). All the hematologic analyses were the same at all four sites”. (Ref: Stavelin A, Flesche K, Tollaanes M, Christensen NG, Sandberg S. Performance of Afinion HbA1c measurements in general practice as judged by external quality assurance data. Clinical Chemistry and Laboratory Medicine (CCLM). 2020;58(4):588-96).

5) Please indicate that you use the Norway version of mentioned scales or the original one. Also, if you used the Norway version, please indicate the reliability and validity of each.

Re: Thank you for raising this concern. The information has now been added to the Methods section for all the PRO questionnaires (page 10-12, lines 218-221, 230-233, 235-243, 249-251, 254-256, 261-267). The following references are added to the reference list. (Ref: Rabin R, Charro Fd. EQ-SD: a measure of health status from the EuroQol Group. Ann Med. 2001;33(5):337-43, Garratt AM, Hansen TM, Augestad LA, Rand K, Stavem K. Norwegian population norms for the EQ-5D-5L: results from a general population survey. Qual Life Res. 2021:1-10, Hanestad BR, Rustoen T, Knudsen O, Lerdal A, Wahl AK. Psychometric properties of the WHOQOL-BREF questionnaire for the Norwegian general population. J Nurs Meas. 2005;12(2):147, and Kalfoss MH, Reidunsdatter RJ, Klöckner CA, Nilsen M. Validation of the WHOQOL-Bref: Psychometric properties and normative data for the Norwegian general population. Health and Quality of Life Outcomes. 2021;19(1):1-12)).

Reviewer #2 

The study aimed to evaluate the effects of an empowerment-based interprofessional follow-up intervention in people with T2D in primary care on patient-reported outcomes, biomarkers, and weight, and to explore the experiences of patients attending the intervention. The manuscript could be improved based on the following comments.

1) Line 87 – 89, the sentence is unclear and requires revision such on the enrollment period, intervention period, follow-up period, and end of study.

Re: Thank you for this important comment. We have revised the text to make this clearer (page 4, lines 89-91). We intended to complete the intervention in 2019-2020, but because of logistic problems the enrollment of patients to the intervention was from August 27th, 2019, to April 15th, 2020, and the follow-up consultations with three GSD consultations were then conducted from December 1st, 2019, until end of study June 15th, 2021.

2) Line 91-97, more detail information on the inclusion and exclusion criteria is to be provided.

Re: Thank you for pointing this out. We have included more information on this at pages 4-5: In the initial cross-sectional survey, exclusion criteria were severe co-morbidity (severe cancer, end stage renal disease), major psychiatric disorder (severe depression, bipolar disorder, schizophrenia), recorded cognitive deficiency, and pregnancy (lines 97-104). In addition, we have added the following reference: Hernar I, Graue M, Igland J, Richards DA, Riise HKR, Haugstvedt A, et al. Patient activation in adults attending appointments in general practice: a cross-sectional study. BMC Primary Care. 2023;24(1):144.

3) Line 101-103, the software which was used in the sample size calculation is to be stated.

Re: Sample size calculation was done using the power-command in Stata. We have added this information at page 5, line 107.

4) Line 110 – what about the control group?

Re: The interview guide included topics such as experiences with participation in the intervention and whether the conversation with the nurses and physicians had led to any change in their ability to manage their health and health care, setting and achieving health goals. Although of general interest, we did not invite participant from the control group to participate in individual interviews. However, this matter might be of interest to the reader, and we have included the following sentence in the limitation section (pages 26-27, lines 517-520): “It might be a limitation that we did not collect data from qualitative interviews with participants in the control group as such data might have shed light on everyday life during the pandemic also for this group”.

5) The effect size index could be presented.

Re: Thank you for questioning this. We assume that the reviewer refers to Cohen’s d , partial eta squared or other similar measures of standardized effect sizes. Unfortunately, there is no established way of calculating such measures based on results from linear mixed effects models, especially not when interaction terms are involved. Due to the way that variance is partitioned in linear mixed models (ref: Rights & Sterba,2019), there does not exist an agreed upon way to calculate standardized effect sizes. Also, it has been recommended that measures of effect should be reported on the scale of measurement, to ease interpretation (ref: Pek & Flora 2018). We therefore believe it would be better to report effect sizes in terms of regression coefficients, which can be interpreted on the same scale as the measurement was done (Ref: Rights, J. D., & Sterba, S. K. (2019). Quantifying explained variance in multilevel models: An integrative framework for defining R-squared measures. Psychological Methods, 24(3), 309–338. https://doi.org/10.1037/met0000184, and Pek, J., & Flora, D. B. (2018). Reporting effect sizes in original psychological research: A discussion and tutorial. Psychological Methods, 23(2), 208–225. https://doi.org/10.1037/met0000126).

6) Line 239, analysis is to be written as Statistical Analysis.

Re: To make this heading clearer we have added two subheadings, page 12, line 287: “Statistical analysis” and page 13, line 310: “Qualitative analysis”.

7) Line 251-252, the sentence unclear and requires revision.

Re: We have revised this to make it clearer (page 12, line 299-301). 

8) Any missing data is to be highlighted including the percentage, pattern etc. The percentage figures stated in the text is to be at least one decimal point.

Re: We have highlighted missing data in table 2 (page 19) and included one decimal point in the text (page 14-15, lines 334-335).

9) Table 1, n for Total could be replaced with N. There are discrepancies in the sample size. If there are missing data or not presented, it is to be presented/highlighted.

Re: Thank you for pointing out this. We have replaced n for total with N and included information on missing data as a footnote in Table 1 (pages 16-17, lines 342-344). 

10) Table 2, statistical test and B is to be denoted in table footnote. If asterisk is to be used in the footnote, the asterisk symbol is to be indicated in the table e.g. B with superscript asterisk. A description of n at 3 months, 6 months and 12 months is to be stated in the table footnote and any missing data is to be denoted.

Re: We have added n columns (at 3 months, 6 months and 12 months) to visualize the number of missing data for every variable in Table 2 (page 19).

11) Line 310, it is not SI Fig 1 but supplementary Table 3. Need to revise the cited Table in Line 296 and Line 314. The supplementary tables numbering in the attachments need to be properly numbered and aligned to the cited table in the text.

Re: Thank you for this information. This has now been checked and appropriately corrected, lines 354, 351 and 368. 

12) For the qualitative interview, it would be good to cite the participants who gave the same opinions/statements so that we know all the 9 participants were involved in the discussion.

Re: In reporting thematic analysis the researchers are not encouraged to focus on frequency or quantify qualitative data in any way for several reasons. In qualitative research, reporting of frequency and the use of simple counts do not tell the story. It is not determined by how many people said it (Ref: Braun, V., & Clarke, V. (2022). Thematic Analysis; a practical guide. SAGE. p. 141-142).

13) Figure 1 and Line 469, the loss of follow-up and reason is to be indicated/stated in the figure.

Re: The loss of participants is shown by numbers in the “boxes”. In the control group, one patient did not participate at 3 months, two patients did not participate at 6 months, however, at 12 months we did not lose these two patients as one of them came back. The person lost in the intervention group participated neither at 3, 6 nor 12 months. Unfortunately, due to ethical and legal regulations the Ethical committee did not allow us to ask for reasons for not remaining in the trial. 

14) Figure 3 is blur, small font and hard to visualize.

Re: Thank you for pointing this out, the figure has been redrawn to make it clearer. 

15) CONSERVE Checklists incomplete.

Re: Thank you for this comment. We have revised accordingly.

16) For CONSORT Checklist, some page numbers are not tally with the manuscript page numbers.

Re: Thank you for this comment. We have corrected this. 

Reviewer #3

1) The authors stated in lines 96-97: “79 consented to participate: 40 people in the intervention group and 36 in the control group.” What happened to the other three individuals who were consented?

Re: We have revised the text to make this clearer (page 5): “Among the identified people with T2D, 79 consented to participate. However, three withdrew consent after randomization leaving 40 people in the intervention group and 36 in the control group (Fig 1.) (lines 101-104). Unfortunately, due to ethical and legal regulations the Ethical committee did not allow us to ask for reasons for the three individuals who had consented but withdraw consent after randomization. 

2) How did the authors check for fidelity? Were the counseling sessions recorded? How do you confirm the counseling sessions were standardized across all participants?

Re: Thank you for an important question that needs clarification. To ensure fidelity in using the GSD method the nurses filled out a predefined reflection sheet after every fifth conversation focusing on how they used the communication skills (mirroring, active listening, and values clarification response). These reflections were discussed with the GSD supervisor (BCHK). This had been added to the text at page 8, lines 178-181. In addition, to further ensure fidelity in applying the GSD, the principles were included in the medical report to standardize its use across all participants during the consultations. This matter has further been clarified at page 7, lines 158-160. 

3) Please explain how you chose the covariates for adjustment. Did you use variable selection in your analysis?

Re: Since this was a randomized trial we did not adjust for any covariates. We only adjusted for the baseline value of the outcome in each model.

4) What time point HbA1c was used for your analysis? how did you handle the scenario if there were multiple HbA1c records within this time period?

Re: We measured HbA1c at baseline and the three follow-up consultations. None of the patients had any other HbA1c measurements. 

5) The authors stated, “There is significant evidence that smaller HbA1c improvements are likely to reduce the risk of diabetes complications.” Can the authors provide more citation with regards to this statement?

Re: Thank you for this comment. We have added two more citations to further illuminate this matter (page 25) (Ref: Ray KK, Seshasai SRK, Wijesuriya S, Sivakumaran R, Nethercott S, Preiss D, et al. Effect of intensive control of glucose on cardiovascular outcomes and death in patients with diabetes mellitus: a meta-analysis of randomised controlled trials. The Lancet. 2009;373(9677):1765-72, and Zoungas S, Arima H, Gerstein HC, Holman RR, Woodward M, Reaven P, et al. Effects of intensive glucose control on microvascular outcomes in patients with type 2 diabetes: a meta-analysis of individual participant data from randomised controlled trials. The lancet Diabetes & endocrinology. 2017;5(6):431-7).

6) This study with a sample size of 76 is very underpowered and the effects seen are marginally significant. How clinically significant are these results?

Re: Thank you for this important comment. We agree that the study was underpowered, and that the effects seen were marginally significant. However, behavioral change takes time and smaller changes might have represented a positive start. This is illuminated by the qualitative data that showed that although only minor changes in health outcomes were captured the intervention opened patients’ eyes to reflection, entailing a greater awareness of health challenges. A longer intervention period of for instance 2-5 years might have said more on this matter. Still, concerns have been made from the perspective of primary care practitioners on the mismatch between resources needed to facilitate access to effective obesity diagnosis and treatments and the organizational structure and capacity of practices. We have added these considerations at pages 24-25, lines 468-472 along with a reference. (Ref: Blane DN, Macdonald S, Morrison D, O’Donnell CA. The role of primary care in adult weight management: qualitative interviews with key stakeholders in weight management services. BMC Health Serv Res. 2017;17(1):1-9).

7) The authors stated in lines 284-285: “We invited nine patients from all four sites to participate in the qualitative interviews and eight consented; four male and four female, mean age 66.8 years (range 55-78 years).” What was the criteria used to invite participants for the interviews? Is it a convenient sampling? What are the associated demerits or limitations?

Re: Thank you for pointing this out. The criteria we used to invite participants were to include participants from the intervention group from all sites and with a balanced representation of both sexes. A variation in age was an additional requirement. It might be a limitation that we did not collect data from qualitative interviews with participants in the control group as such data might have shed light on everyday life under the pandemic also for this group. We have added these reflections to the Methods section and the limitations (pages 5 and 26-27, lines 119-120 and 517-520).

In addition to the changes to the manuscript based on the constructive comments from the reviewers, also some grammatical errors have been corrected.

---

## [Editor Report · Decision Letter 1]

25 Aug 2023

Evaluation of an interprofessional follow-up intervention among people with type 2 diabetes in primary care – a randomized controlled trial with embedded qualitative interviews.

PONE-D-23-11188R1

Dear Dr. Graue,

We’re pleased to inform you that your manuscript has been judged scientifically suitable for publication and will be formally accepted for publication once it meets all outstanding technical requirements.

Kind regards,

Jennifer Annette Campbell, PhD, MPH

Academic Editor

PLOS ONE
---

## [Editor Report · Acceptance letter]

7 Sep 2023

PONE-D-23-11188R1 

Evaluation of an interprofessional follow-up intervention among people with type 2 diabetes in primary care – a randomized controlled trial with embedded qualitative interviews. 

Dear Dr. Graue:

I'm pleased to inform you that your manuscript has been deemed suitable for publication in PLOS ONE. Congratulations! Your manuscript is now with our production department. 

Kind regards, 

on behalf of

Dr. Jennifer Annette Campbell 

Academic Editor

PLOS ONE